# Clinical Variables Related to Functional Capacity and Exertional Desaturation in Patients with COVID-19

**DOI:** 10.3390/biomedicines11072051

**Published:** 2023-07-20

**Authors:** Santiago Larrateguy, Julian Vinagre, Federico Londero, Johana Dabin, Evangelina Ricciardi, Santiago Jeanpaul, Rodrigo Torres-Castro, Rodrigo Núñez-Cortés, Diana Sánchez-Ramírez, Elena Gimeno-Santos, Isabel Blanco

**Affiliations:** 1Servicio de Kinesiología y Fisioterapia, Hospital de la Baxada “Dra. Teresa Ratto”, Paraná 3100, Argentina; santilarra@gmail.com (S.L.); julianvinagre10@gmail.com (J.V.); londero2@hotmail.com (F.L.); jdabindittler@gmail.com (J.D.); mericciardi85@gmail.com (E.R.); santiagojeanpaul@gmail.com (S.J.); 2Facultad de Ciencias de la Salud, Universidad Adventista del Plata, Libertador San Martin 3103, Argentina; 3Department of Physical Therapy, Faculty of Medicine, University of Chile, Santiago 8380453, Chile; rodritorres@uchile.cl (R.T.-C.); r_nunez@uchile.cl (R.N.-C.); 4Department of Pulmonary Medicine, Hospital Clínic, University of Barcelona, 08036 Barcelona, Spain; iblanco2@clinic.cat; 5Institut d’Investigacions Biomèdiques August Pi i Sunyer (IDIBAPS), 08036 Barcelona, Spain; 6Physiotherapy in Motion Multispeciality Research Group (PTinMOTION), Department of Physiotherapy, University of Valencia, 46003 Valencia, Spain; 7Department of Respiratory Therapy, College of Rehabilitation Sciences, University of Manitoba, Winnipeg, MB R3T 2N2, Canada; diana.sanchez-ramirez@umanitoba.ca; 8Biomedical Research Networking Center on Respiratory Diseases (CIBERES), 30627 Madrid, Spain; 9Barcelona Institute for Global Health (ISGlobal), 08036 Barcelona, Spain

**Keywords:** COVID-19, SARS-CoV-2, coronavirus disease 2019, exercise capacity, rehabilitation

## Abstract

Impaired functional capacity is one of the most commonly reported consequences among post-COVID-19 patients. This study aimed to analyse the clinical variables related to functional capacity and exertional desaturation in post-COVID-19 patients at the time of hospital discharge. A cross-sectional study was conducted on patients recovering from COVID-19 pneumonia. The main outcomes measures were functional capacity, assessed using the 1 min sit-to-stand test (1 min STST), and exertional desaturation, defined as a drop of ≥4% in the arterial oxygen saturation. Factors used to characterise the participant outcomes included the use of a high-flow nasal cannula (HFNC), prolonged hospitalisation, occurrence of pulmonary embolism during hospitalisation, and underlying comorbidities. A total of 381 participants (mean age = 53.7 ± 13.2 years, 65.6% men) were included. Participants completed a mean of 16.9 ± 6.2 repetitions in the 1 min STST. Exertional desaturation was observed in 51% of the patients. Higher odds of exertional desaturation were found in the participants who used a HFNC (OR = 3.6; 95%CI: 1.6 to 7.8), were admitted in the hospital >10 days (OR = 4.2; 95%CI: 2.6 to 6.8), and had a pulmonary embolism (OR = 3.5; 95%CI: 2.2. to 5.3). Use of a HFNC (β = −3.4; 95%CI: −5.3 to −1.44), a hospital stay >10 days (β = −2.2; 95%CI: −3.4 to −0.9), and a history of pulmonary embolism (β = −1.4; 95%CI: −2.6 to −0.2) were also negatively associated with the 1 min STST. Most post-COVID-19 patients exhibited reduced functional capacity at the time of hospital discharge, and approximately half had exertional desaturation after the 1 min STST. The use of a HFNC, prolonged hospitalisation and pulmonary embolism were the main clinical variables associated with worse a 1 min STST performance and a higher likelihood of exertional desaturation.

## 1. Introduction

The coronavirus disease (COVID-19) has been a challenge for health systems across the world, affecting more than 670 million people, with more than 6.8 million deaths by May 2023 [1]. Although the majority of people infected by the severe acute respiratory syndrome coronavirus 2 (SARS-CoV-2) developed asymptomatic or mild disease, about 20% developed severe disease requiring hospitalisation, and close to 6% required critical care in an intensive care unit [2]. Among the severe cases, pulmonary embolism is a frequent complication associated with the clinical worsening of COVID-19 [3]. In addition, some cases may require respiratory support (e.g., a high-flow nasal cannula (HFNC)) for the treatment of acute hypoxemic respiratory failure [4]. Thus, prolonged hospitalisation due to COVID-19 complications may lead to worse outcomes at discharge. Although COVID-19 is primarily a respiratory disease, it can affect multiple systems, such as the cardiovascular or neurological, leaving a vast number of sequelae that impact the patient’s quality of life and the ability to return to work [5,6,7,8]. Among the most reported sequelae are fatigue, dyspnoea, and impairment of functional capacity [6,9].

Due to the functional limitations that COVID-19 generates in a significant part of the population, different national health systems have developed follow-up programmes focused on imaging, lung function, symptoms, and functional capacity [10,11,12]. One of the pillars of the follow-up and intervention programmes is the evaluation of functional capacity [13], which can be assessed with laboratory tests, such as the cardiopulmonary exercise test (CPET) or field tests, such as the six-minute walk test (6MWT) or the 1 min sit-to-stand test (1 min STST) [14,15,16,17,18].

The 6MWT is the most commonly used test for respiratory, cardiological, metabolic, or neurological diseases [14]. This test has been widely demonstrated to be helpful in assessing functional capacity and can be performed in low-resource contexts [14]. However, to provide specific information about functional or exercise capacity, a test should be chosen according to the characteristics of each subject, the setting, and the physiologically expected answer [19]. The 6MWT requires a 30 m corridor (at least 20 m), which is often unavailable in hospitals or rehabilitation centers and even less at home [14]. The 1 min STST has the advantage of requiring a small space compared to the 6MWT, and less sophisticated equipment as compared with tests using treadmills or cycle ergometers; as such it may be an alternative to evaluate functional capacity when the 6MWT cannot be performed [16]. The 1 min STST has significantly correlated with the 6MWT in patients with different diseases, including post-COVID-19, chronic obstructive pulmonary disease (COPD), and pulmonary hypertension (PH) patients [16,20,21]. 

Functional capacity assessments are widely used in intervention programmes such as pulmonary rehabilitation [22,23,24]. Due to the high number of patients left with sequelae and the recommendation to evaluate functional capacity according to the guidelines recommendations [12,25], it is necessary to determine which clinical variables may affect the results of functional evaluations and to identify the people with an increased risk of having a poor result. Therefore, our objective was to analyse the clinical variables related to functional capacity and exertional desaturation in post-COVID-19 patients at hospital discharge.

## 2. Materials and Methods

### 2.1. Design and Participants

We conducted a cross-sectional study in patients recovering from COVID-19 pneumonia once they were discharged from the Hospital de la Baxada between April 2021 and March 2022. Ethics committee approval was obtained, and all patients signed the informed consent. This study followed the recommendations of the STrengthening the Reporting of OBservational studies in Epidemiology guidelines (STROBE) [26].

The inclusion criteria were as follows: patients older than 18 years, and a diagnostic of COVID-19 by positive PCR assay findings for nasal and pharyngeal swab specimens. In addition, the exclusion criteria were the presence of locomotor or cognitive impairment before the infection, refusal to participate, and any pre-existing condition, such as orthopaedic or neurological conditions, that limited the ability to perform the 1 min STST. 

### 2.2. Measurements

Demographic characteristics, medical history, exposure history, and underlying comorbidities were collected at discharge. The main outcome measure was functional capacity, assessed through the 1 min STST at hospital discharge. All tests were conducted in the same room, with only the presence of the evaluator and the patient, to avoid distractions. 

The 1 min STST was performed with a standard height chair (46 cm) without armrests, positioned against a wall. Participants were not allowed to use their hands/arms to push the chair’s seat or their body. Participants were instructed to complete as many sit-and-stand cycles as possible in 60 s at a self-paced speed [25]. We used the reference values based on the healthy adult population previously reported by Strassmann et al. [27].

A finger oximeter was used to record the oxygen saturation (SpO_2_) and heart rate (HR). A drop of ≥4% in the arterial oxygen saturation was considered clinically significant [28]. The evaluator had previous experience (5 years) in performing field tests to assess physical capacity, including the 6MWT and 1 min STST. The 6MWT was performed following the recommendations of the European Respiratory Society/American Thoracic Society (ERS/ATS) clinical guidelines [14]. The 1 min STST was performed only once to avoid a learning effect [29].

The clinical variables were as follows: (1) Use of a HFNC during hospitalisation; (2) prolonged hospitalisation. Based on previous studies in patients with COVID-19, length of stay >10 days was established as the cut-off point for defining a prolonged hospital stay [18,30]; (3) pulmonary embolism during hospitalisation; (4) history of underlying comorbidities (diabetes, hypertension, or chronic respiratory disease) at the time of hospitalization; (5) obesity (i.e., body mass index ≥ 30 kg/m^2^)

### 2.3. Statistics

All statistical analyses were performed with SPSS software (v. 22.00 for Windows, Chicago, IL, USA). The normality of the data distribution was assessed using the Shapiro–Wilk test. Data were described as the mean ± standard deviation, frequency, and percentages. To evaluate the associations of each of the clinical variables with the exertional desaturation after the 1 min STST (outcome), binary logistic regression analysis adjusted for sex and age (covariates) was performed. Data were presented as the odds ratio (OR) with 95% confidence intervals (95%CI). The effect sizes of the OR were characterised as small, moderate, or large and were established by an OR of 1.68, 3.47, and 6.71, respectively [31]. The sample size was pragmatic and depended on the ability of the clinical staff to recruit the participants continuously and to collect the data.

To evaluate the associations of each of the clinical variables with the number of repetitions in the 1 min STST (outcome), a simple linear regression analysis adjusted for sex and age (covariates) was performed. Data were presented as the regression coefficient (β), with 95%CI. The level of statistical significance was set at *p* < 0.05.

## 3. Results

A total of 381 participants (mean age = 53.7 ± 13.2 years, 65.6% men) were included in the study (Table 1). The mean number of repetitions in the 1 min STST was 16.9 ± 6.2. Moreover, 78.4% of the cases obtained results below the lower limit of normality (percentile 2.5), according to the reference values. A total of 51.1% of the patients presented exertional desaturation after the 1 min STST. Table 2 shows the adjusted models for the association between the clinical variables and exertional desaturation after the 1 min STST. Participants with a history of HFNC (OR = 3.6; 95%CI = 1.6 to 7.8), hospital stay >10 days (OR = 4.2; 95%CI = 2.6 to 6.8), and pulmonary embolism (OR = 3.5; 95%CI = 2.2. to 5.3) had a significantly higher risk of exertional desaturation, with a moderate effect size. Table 3 shows the adjusted models for the association between the clinical variables and functional capacity. Use of a HFNC (β = −3.4; 95%CI = −5.3 to −1.44]), hospital stay >10 days (β = −2.2; 95%CI = −3.4 to −0.9) and a history of pulmonary embolism (β = −1.4; 95%CI = −2.6 to −0.2) were negatively associated with the number of repetitions in the 1 min STST.

## 4. Discussion

At hospital discharge, most patients that had recovered from acute COVID-19 infection had decreased functional capacity, and approximately half had exertional desaturation. The patients with a history of HFNC, prolonged hospitalisation, and pulmonary embolism, had worse 1 min STST performance and a higher likelihood of exertional desaturation. We found that almost 80% of the patients had a functional capacity lower than the 2.5th percentile of the reference values used. Our results were in line with other reports that showed a great affectation in the functional capacity of post-COVID-19 patients [18,24].

Our findings showed that hospitalisation for more than ten days and using respiratory support (through the HFNC) increased the risk of exertional desaturation. These were not unexpected since the most severe patients require ventilatory support and consequently spend more days hospitalised in critical units [32,33]. Nevertheless, the literature has shown that impaired functional capacity is not necessarily related to the severity of the disease [34]; damage caused by the virus or generated by ventilatory dependence must also be considered together with the harmful effects of prolonged rest [33].

The patients who had a pulmonary embolism during hospitalisation had an OR of 3.5 to develop exertional desaturation. This aligned with a recent study that found that 29% (24/84) of patients with COVID-19 had a pulmonary embolism; the authors reported a lower level of peripheral oxygen saturation (86.8% vs. 88.6% *p* = 0.016) and longer time of hospitalisation (*p* < 0.01) in patients with a pulmonary embolism compared with the no-pulmonary embolism cases [35]. These findings were related to structural damage, as in 87% of patients, the pulmonary embolism was found in the lung parenchyma affected by COVID-19 pneumonia, with a worse chest tomography severity score and a greater number of lung lobar involvement compared with the non-pulmonary embolism patients [35].

The pathogenesis of COVID-19 associated with pulmonary embolism is unclear [35]. However, it has been reported that some patients with COVID-19 showed pulmonary vascular compromise [36]. On the other hand, some studies have reported vascular compromise in areas of pulmonary opacities, which could indicate an inflammatory response with vascular involvement leading to thrombosis [37,38].

The use of the 1 min STST proved to be a good tool to assess functional capacity and exertional desaturation in the post-COVID-19 patients, which, given its advantages in terms of low space and equipment requirements, could be applied in different settings (e.g., in the office or in telehealth) to identify cases with major functional limitations, and to guide rehabilitation teams in decision making (e.g., exercise prescription) [39]. In fact, the 30 s sit-stand test, as a variant of the sit-to-stand test, has been shown to be a viable and safe option for telehealth assessment and is associated with persistent post-COVID-19 sequelae (e.g., fatigue, dyspnoea, and pain) in non-hospitalised patients [40]. Although there are no protocols comparing the 1 min STST or the 30s STST in patients with COVID-19, there are studies that compare them in COPD showing that the 1 min STST was even better associated with important clinical outcomes such as functional exercise capacity, functional status, and physical activity in daily life [41]. Consequently, healthcare professionals may use this method (as well as the 1 min STST) when face-to-face assessment of physical COVID-19 sequelae is not feasible due to geographical and socio-economic constraints. 

On the other hand, this assessment should be complemented by evaluating other relevant health indicators in post-COVID-19 patients, such as pulmonary function [8], social factors [42], comorbidity burden [43], performance in activities of daily living [44], and persistent symptoms such as dyspnea and fatigue [40]. In addition, it should be noted that there were several factors, both from the patients’ and the testing centre’s perspective, that influenced the patients’ physical capacity. For example, advanced age and frailty were strongly associated with reduced functional capacity in COVID-19 survivors [45,46]. These factors can affect muscle strength (older people take longer to regain muscle strength), mobility, endurance, and balance, which may affect independence and quality of life [47]. Therefore, physical function tests such as the short physical performance battery are recommended for this population [48]. Also, cognitive status, which may be impaired in post-ICU patients due to medication use and/or the presence of delirium [13], may have limited the assessment of physical capacity by using a test that required the following of instructions [14]. Therefore, the characteristics of each individual must be considered when selecting the appropriate test.

Our assessment was mainly based on evaluating physical capacity. However, the long-term effects on physical capacity are closely associated with lung damage, which can be measured through the lung function test and imaging evaluations. The existing literature indicates that approximately 50% of patients exhibit residual lung function abnormalities three months after hospital discharge [49], which is higher in the post-ICU patients [50]. We did not perform lung function evaluation, since clinical guidelines suggest an evaluation between 8- and 12-weeks post discharge [12], since the lung function would reflect the exaggerated inflammatory response of the host to viral pneumonia with severe gas exchange impairment and excessive stress and strain on the lung parenchyma [50,51] more than the actual lung function of the patients.

### Strengths and Limitations

The strengths of our study included the relatively large sample size and the use of a validated tool to assess the main outcome. In contrast, this study had some limitations. First, due to the nature of the cross-sectional analysis, we could not establish a causal relationship between the clinical variables and outcomes. Therefore, the results should be interpreted with caution. In addition, we could not rule out selection bias due to the nature of convenience sampling. Also, the absence of radiological images as well as the underreporting of comorbidities in each patient’s medical history may have underestimated the true strength of the association of these variables with the outcomes. On the other hand, information on the level of physical activity prior to hospitalisation was unavailable. Therefore, residual confounding bias was possible due to a lack of adjustment for physical activity variables. Finally, the results were compared with international reference values since they are unavailable for our country.

## 5. Conclusions

Most post-COVID-19 patients experienced a decrease in functional capacity at the time of hospital discharge, and approximately half of them exhibited exertional desaturation after the 1 min STS. The use of a HFNC, prolonged hospitalization, and the occurrence of a pulmonary embolism during hospitalisation were the main clinical variables associated with poorer performance in the 1 min STST performance and with a higher likelihood of exertional desaturation. Future studies should assess the energy expenditure and oxygen consumption to determine if, from a metabolic standpoint, the 1 min STST behaves similarly to the 6MWT in post-COVID-19 patients. Clinical guidelines should incorporate this type of field test to facilitate the evaluation of physical capacity and exercise-induced desaturation, with a focus on patients with a history of HFNC use, prolonged hospitalisation, and pulmonary embolism.

## Figures and Tables

**Table 1 biomedicines-11-02051-t001:** Clinical characteristics of patients (*n* = 381).

Characteristics	Value
Age (years)	53.7 ± 13.2
Sex male *n* (%)	250 (65.6)
BMI (kg/m^2^)	31.3 ± 6.3
HFNC therapy, *n* (%)	39 (10.3)
Hospital stay (days)	9.5 ± 6.6
Hospital stay >10 days, *n* (%)	124 (32.5)
Pulmonary embolism, *n* (%)	161 (42.2)
Diabetes, *n* (%)	90 (23.6)
Hypertension, *n* (%)	141 (37.0)
Chronic respiratory disease, *n* (%)	21 (5.5)
Obesity, *n* (%)	200 (52.4)
1 min STST (repetitions)	16.9 ± 6.2
Repetitions < 2.5th percentile, *n* (%)	299 (78.4)
Exertional desaturation, *n* (%)	195 (51.1)

Data are presented as mean ± standard deviation or *n* (%). Abbreviations: BMI = Body mass index; HFNC = High-flow nasal cannula; 1 min STST = 1 min sit-to-stand test.

**Table 2 biomedicines-11-02051-t002:** Associations between clinical variables and exertional desaturation after 1 min STST in post-COVID-19 patients.

Clinical Variables	Category	Adjusted OR [95%CI] ^a^
HFNC therapy	Yes vs. No	**3.6 [1.6 to 7.8]**
Hospital stay > 10 days	Yes vs. No	**4.2 [2.6 to 6.8]**
Pulmonary embolism	Yes vs. No	**3.5 [2.2. to 5.3]**
Diabetes	Yes vs. No	1.0 [0.6 to 1.6]
Hypertension	Yes vs. No	1.1 [0.7 to 1.8]
Chronic respiratory disease	Yes vs. No	0.9 [0.4 to 2.1]
Obesity	Yes vs. No	**1.6 [1.1 to 2.4]**

Abbreviations: CI = confidence interval; HFNC = High-flow nasal cannula. ^a^ Adjusted for sex and age.

**Table 3 biomedicines-11-02051-t003:** Associations between clinical variables and number of repetitions in the 1 min STST in post-COVID-19 patients.

Clinical Variables	Category	Adjusted β [95%CI] ^a^
HFNC therapy	Yes vs. No	**−3.4 [−5.3 to −1.44]**
Hospital stay > 10 days	Yes vs. No	**−2.2 [−3.4 to −0.9]**
Pulmonary embolism	Yes vs. No	**−1.4 [−2.6 to −0.2]**
Diabetes	Yes vs. No	−0.3 [−1.7 to 1.2]
Hypertension	Yes vs. No	−0.4 [−1.7 to 0.9]
Chronic respiratory disease	Yes vs. No	−1.7 [−4.3 to 0.9]
Obesity	Yes vs. No	0.4 [−0.9 to 1.6]

Abbreviations: CI = confidence interval; HFNC = High-flow nasal cannula. ^a^ Adjusted for sex and age.

## Data Availability

The data presented in this study are available on reasonable request from the corresponding author.

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
