# Peer review of "Clinical Variables Related to Functional Capacity and Exertional Desaturation in Patients with COVID-19"

_biomedicines, 2023, doi:10.3390/biomedicines11072051_

Round 1

Reviewer 1 Report

The study is very interesting and conducted with good scientific rigor.

It would also be interesting to compare the data produced with imaging data.

It would also be useful to compare the results already documented in the literature, examples of which I report:

Orzes N, Pini L, Levi G, Uccelli S, Cettolo F, Tantucci C. A prospective evaluation of lung function at three and six months in patients with previous SARS-COV-2 pneumonia. Respir Med. 2021 Sep;186:106541. Doi: 10.1016/j.rmed.2021.106541. Epub 2021 Jul 10. PMID: 34280885; PMCID: PMC8272067.

Pini L, Montori R, Giordani J, Guerini M, Orzes N, Ciarfaglia M, Arici M, Cappelli C, Piva S, Latronico N, Muiesan ML, Tantucci C. Assessment of respiratory function and exercise tolerance at 4-6 months after COVID-19 infection in patients with pneumonia of different severity. Intern Med J. 2023 Feb;53(2):202-208. doi: 10.1111/imj.15935. Epub 2022 Sep 28. PMID: 36114661; PMCID: PMC9538800.

The English should be thoroughly revised.

Author Response

The responses are attached.

Reviewer 2 Report

Thank you for the opportunity to review this article.

wonder if COVID-19 should be key word

line 30 should start sentence with word so three hundred or turn sentnece around, same line 32

line 54 paragrapghs shoud be more the two sentences

line 65 is there any evidence of the validity of the STST test and equivalency - needs to be oncluded here

line 108 what test - if this is STST then maybe some discussion about the expereince

line 109 what test

results - start with main points of demographic data and then refer to table. The discussion of on STST and table. The discussion of what is in table. Better to have table close to text where it is being discussed

ths section needs developing more as well

line 155 could combine these two paragraphs as each para is only two sentences whihc is not a para

line 184 should be new sentence as this sentence is too long and not clear

line 187 not clear what the difference between these two are

line 198 plus older people take longer to regain muscle strength

line 208 not provided evidence of tool having been validated

conclsuion lacks the 'so what' and needs developing. What about recommendatins. Should there be some work comparing these two tests and where to next

Wish you well with your ongoing research

language good, only minor issues

Author Response

The responses are attached

Reviewer 3 Report

The title of the manuscript is concise, informative.

English language has high quality.

The tables meet the required standards.

The introduction of the manuscript is well-organized and effectively emphasizes the importance of the research topic.

The materials and methods section provides a thorough and detailed description.

In the results section, the findings are presented clearly and concisely.

The discussion showcases a strong comprehension of the results and their implications.

The authors adeptly connect their findings to previous research and offer insightful interpretations and potential explanations for the observed outcomes.

The strengths and limitations were outlined by authors.

The conclusion provides a brief summary of the main findings and their significance in relation to the research question.

Author Response

Thank you very much for your good comments.